# TRAF4 Promotes the Proliferation of Glioblastoma by Stabilizing SETDB1 to Activate the AKT Pathway

**DOI:** 10.3390/ijms231710161

**Published:** 2022-09-05

**Authors:** Hongyu Gu, Shunqin Zhu, Cheng Peng, Zekun Wei, Yang Shen, Chaoyu Yuan, He Yang, Hongjuan Cui, Liqun Yang

**Affiliations:** 1State Key Laboratory of Silkworm Genome Biology, Southwest University, Chongqing 400715, China; 2Cancer Center, Medical Research Institute, Southwest University, Chongqing 400715, China; 3School of Life Sciences, Southwest University, Chongqing 400715, China

**Keywords:** glioblastoma, TRAF4, SETDB1, AKT, ubiquitination

## Abstract

The process of ubiquitination regulates the degradation, transport, interaction, and stabilization of substrate proteins, and is crucial for cell signal transduction and function. TNF receptor-associated factor 4, TRAF4, is a member of the TRAF family and is involved in the process of ubiquitination as an E3 ubiquitin protein ligase. Here, we found that TRAF4 expression correlates with glioma subtype and grade, and that TRAF4 is significantly overexpressed in glioblastoma and predicts poor prognosis. Knockdown of TRAF4 significantly inhibited the growth, proliferation, migration, and invasion of glioblastoma cells. Mechanistically, we found that TRAF4 only interacts with the Tudor domain of the AKT pathway activator SETDB1. TRAF4 mediates the atypical ubiquitination of SETDB1 to maintain its stability and function, thereby promoting the activation of the AKT pathway. Restoring SETDB1 expression in TRAF4 knockdown glioblastoma cells partially restored cell growth and proliferation. Collectively, our findings reveal a novel mechanism by which TRAF4 mediates AKT pathway activation, suggesting that TRAF4 may serve as a biomarker and promising therapeutic target for glioblastoma.

## 1. Introduction

Gliomas are the most common primary malignant central nervous system (CNS) tumors and carry a very poor prognosis. Glioblastoma multiforme (GBM) is a grade IV glioma, accounting for 57% of all gliomas and 48% of CNS primary malignancies [1,2]. A large-scale clinical study on glioblastoma showed that the age-adjusted incidence rate of glioblastoma was 3.21 cases/100,000 people. The median age of patients with glioblastoma is 65 years, and the incidence increases with age, peaking at 75–84 years of age [3]. Advances in basic and clinical medicine have significantly improved the short-term survival expectations of glioblastoma patients. However, the long-term survival of glioblastoma patients, with a 5-year survival rate of 5.8%, is still quite low [1,4,5]. These have forced us to pay more attention to biomarkers, mechanisms of progression, and promising medical treatments of glioblastoma, such as immunotherapy and targeted therapy.

In eukaryotic cells, the TNF receptor-associated factor (TRAF) family has seven members, from TRAF1 to TRAF7 [6]. They play important roles in receptor-mediated cell signaling and ubiquitination modification [7,8,9]. TRAF4, an important member of the TRAF family, was initially identified as a highly expressed differential gene in breast cancer [10]. TRAF4 has nuclear localization signals (NLS) which distinguish it from other family members; therefore, TRAF4 is distributed in the cell membrane, cytoplasm, and nucleus [10,11]. TRAF4 is involved in various important signal transduction in cells and it plays important functions in the immune system, nervous system, tumor growth and metastasis, cell differentiation, and DNA damage response [12,13,14,15,16].

Ubiquitination regulation is fundamental to cell function and signal communication. Existing studies have shown that three enzymes are involved in the ubiquitin cascade, namely, E1 ubiquitin activating enzyme, E2 ubiquitin conjugating enzyme, and E3 ubiquitin protein ligase [17]. Among them, E3 ubiquitin protein ligase mediates the attachment of ubiquitin proteins to the substrate, which is a key step in the ubiquitin cascade [18]. TRAF4 is a member of the E3 ubiquitin protein ligase, and its N-terminal RING domain exerts the ubiquitin ligation activity. TRAF4 is overexpressed in many tumors notably, and because of its important ubiquitination function, it is involved in key signaling pathways in tumor progression [19,20]. Non-proteolytic ubiquitination of tyrosine receptor kinase A (TrkA) by TRAF4 promotes the kinase activity of TrkA, which stimulates downstream p38 MAPK activation and invasion-associated genes expression in prostate cancer [19]. In colorectal cancer, TRAF4 catalyzes the ubiquitination of Checkpoint Kinase 1 (CHK1), which is required for CHK1 phosphorylation and ATR activation following DNA damage [15]. So far, only few studies have explored the role of TRAF4 in glioblastoma [21,22], but the rather important ubiquitination regulation of TRAF4 has not been investigated. What role does TRAF4 play in the pathogenesis of glioblastoma, and which signaling pathways are involved? These issues still need to be further studied.

Here, we report that TRAF4 is significantly overexpressed in glioblastoma and is important for cell growth, proliferation, migration, and invasion. Mechanistically, TRAF4 interacts with the Tudor domain of histone methyltransferases SET domain bifurcated 1 (SETDB1), regulates the stability of the latter through atypical ubiquitination, and ultimately promotes the activation of the AKT pathway. Our findings clarify the reasons for the high expression of TRAF4 and the abnormal activation of the AKT pathway in glioblastoma. These reveal the ubiquitination role of TRAF4 in glioblastoma and may provide new insights and targets for glioblastoma diagnosis and treatment.

## 2. Results

### 2.1. TRAF4 Is Highly Expressed in Glioblastoma, and High Expression Indicates Poor Prognosis

To explore the role of TRAF4 involved in tumorigenesis, we analyzed the expression of TRAF4 in tumor and adjacent tissues in the GEPIA database. The results show that across multiple cancers, TRAF4 was more highly expressed in most tumors than that in the corresponding paracancerous tissues, including glioblastoma (Figure 1A). To characterize TRAF4 in gliomas, we analyzed data from the TCGA, CGGA, Rembrandt, and Gravendeel database [23]. Consistent with the data from GEPIA, TRAF4 was expressed at higher levels in GBM than that in non-tumor tissues (Figure 1B and Appendix A). The expression of TRAF4 in various histological subtypes of gliomas was further analyzed. TRAF4 expression in glioblastoma was significantly higher than that in other low-malignancy gliomas (Figure 1C and Appendix A). The WHO classifies malignant gliomas into grade II, III, and IV. As the results showed, the expression of TRAF4 increased with the grade of glioma (Figure 1D and Appendix A). Furthermore, the TCGA database also shows that the gene copy number gain for TRAF4 was linked to high expression of TRAF4 in glioma (Figure 1E). To investigate the relationship between TRAF4 expression and prognosis of glioma patients, we analyzed the survival data of glioma patients in the TCGA and CGGA databases. In both databases, we found that glioma patients with high TRAF4 expression had significantly shorter survival time than those with low TRAF4 expression (Figure 1F).

The protein expression of TRAF4 was significantly higher in glioblastoma cell lines compared to human astroglia cell SVGP12 (Figure 1G). We, therefore, selected glioblastoma cell lines U87-MG and LN-229 for subsequent experiments. Collectively, our analysis of glioma databases and Western blot experiments revealed that TRAF4 is overexpressed in glioblastoma and high expression predicts poor prognosis in patients.

### 2.2. Loss of TRAF4 Inhibits the Proliferation of Glioblastoma Cells

To verify the role of TRAF4 in glioblastoma cells, we knocked down the expression of TRAF4 in U87-MG and LN-229 cells using shRNA lentiviral constructs (Figure 2A). The results of MTT experiments showed that knockdown of TRAF4 significantly inhibited the cell proliferation of glioblastoma cells (Figure 2B). 5-ethynyl-2′-deoxyuridine (EdU) incorporation experiments showed that knockdown of TRAF4 reduced DNA synthesis ability and proliferation in glioblastoma cells (Figure 2C). We further examined the growth ability of glioblastoma. The colony formation assays indicated that knockdown of TRAF4 suppressed the colony formation ability of glioblastoma cells (Figure 2D). In the in vitro tumor formation assay, the tumor formation ability of glioblastoma cells following TRAF4 knockdown was significantly decreased (Figure 2E). Wound-healing experiments suggested that TRAF4-knockdown LN-229 cells had reduced metastatic capacity (Appendix A). Transwell assays were employed to detect cell migration and invasion, and the results showed that knockdown of TRAF4 significantly inhibited the migration and invasion of glioblastoma cells (Appendix A). In conclusion, our results suggest that loss of TRAF4 significantly inhibits glioblastoma proliferation.

### 2.3. Knockdown of TRAF4 Inhibits Activation of the AKT Pathway in Glioblastoma, Whereas Restoration of TRAF4 Expression Counteracts the Inhibition

In previous experiments, we found that knockdown of TRAF4 inhibited the proliferation of glioblastoma. Here, first, we examined the expression of genes involved in growth and metastasis in glioblastoma cells. Western blot results showed that the growth-related proteins (CCND1 and β-catenin) of glioblastoma cells decreased following TRAF4 knockdown, consistent with previous studies [22,24,25]. We detected cell metastasis-related proteins. The level of MMP2 protein decreased and the level of E-cadherin protein increased, which verified that the metastasis ability of glioblastoma cells was inhibited, which is consistent with the regulatory mechanism involved in TRAF4 in breast cancer [26] (Figure 3A). To further validate the experimental conclusions, we explored the relationship between TRAF4 and these growth- and metastasis-related genes in the glioma clinical databases TCGA and CGGA. Correlation analysis suggested that TRAF4 was significantly positively correlated with CCND1 and MMP2 in both databases (Appendix A). We downloaded and utilized CGGA clinical data for gene set enrichment analysis (GSEA). The results of GSEA indicated that TRAF4 expression was significantly positively correlated with CCND1-related and vascular endothelial growth factor (VEGF)-related gene sets (Figure 3D). Canonical VEGF signaling is involved in guiding cell proliferation, metastasis, survival, and angiogenesis [27]. Overall, the results of analyzing the database suggested that TRAF4 is involved in the proliferation of glioblastoma, consistent with the findings in our cellular experiments.

TRAF4 mediates activation of the AKT pathway in many cases [28,29]. Here, we wanted to examine whether TRAF4 also mediates the activation of the AKT pathway in glioblastoma. The results of Western blotting showed that knockdown of TRAF4 inhibited the activation of AKT pathway in glioblastoma cells (Figure 3B). The results of GSEA also indicated that TRAF4 expression was significantly positively correlated with many genes in the AKT pathway (Figure 3C).

We continued to dig deeper into the results of the GSEA analysis. Interestingly, we found that high TRAF4 expression was significantly positively correlated with the histone modification gene set and histone lysine methylation gene set (Figure 3E). Among the top-ranked genes in these two gene sets, we found SETDB1, an important factor involved in the activation of the AKT pathway [30,31]. SETDB1 cooperates with phosphoinositide-3-kinase (PI3K) to activate the AKT pathway, promoting membrane recruitment and phosphorylation of AKT. Loss of SETDB1 inhibits AKT kinase activity and oncogenic function [30,31]. The results of Western blot showed that the SETDB1 expression was also decreased following TRAF4 knockdown in glioblastoma cells (Figure 3B). Correlation analysis in TCGA and CGGA glioma clinical databases also suggested that TRAF4 was significantly positively correlated with SETDB1 (Figure 3F and Appendix A). We restored TRAF4 expression in glioblastoma cells following TRAF4 knockdown. The results of Western blot indicated that restoring TRAF4 expression counteracted the inhibitory effect of TRAF4 knockdown on AKT pathway activation (Figure 3G). As the results showed, the SETDB1 expression was also increased following restoring TRAF4 expression in glioblastoma cells (Figure 3G). Collectively, these results strongly suggested that TRAF4 regulated cell proliferation and mediated activation of the AKT pathway in glioblastoma.

### 2.4. TRAF4 Interacts with the Tudor Domain of SETDB1

We hypothesized that TRAF4 and SETDB1 are co-involved in the activation of the AKT pathway in glioblastoma based on previous results. Firstly, we validated the interaction of TRAF4 and SETDB1 with AKT1 in glioblastoma cells. Immunoprecipitation experiments showed that both TRAF4 and SETDB1 interacted with AKT1 in glioblastoma cells (Figure 4A). We performed protein–protein interaction (PPI) analysis in two online databases (GeneMANIA and String), and the results suggested that there may be an interaction between TRAF4 and SETDB1 (Figure 4B and Appendix A). To explore the relationship between TRAF4 and SETDB1, we overexpressed Flag-TRAF4 and MYC-SETDB1 in 293FT cells. We found that exogenously expressed Flag-TRAF4 interacted with MYC-SETDB1 in 293FT cells (Figure 4C). Further immunoprecipitation experiments in glioblastoma cells showed that endogenous TRAF4 likewise interacted with SETDB1 (Figure 4D,E).

The SETDB1 protein mainly has three domains, which are a tandem domain consisting of two Tudors, a CpG DNA methyl binding domain (MBD), and a bifurcated SET domain [32] (Figure 5A). In order to further explore the binding mode of TRAF4 and SETDB1, according to the three domains of SETDB1, we truncated the wild-type SETDB1 sequentially from the C-terminal to the N-terminal. We generated different deletion fragments of SETDB1, one with the SET domain deleted (ΔSET) and the other with only the N-terminal Tudor domain (Tudor) (Figure 5A). We calculated that the molecular weight of the full-length MYC-SETDB1-WT protein is about 170 kDa, the deletion fragment MYC-SETDB1-ΔSET is about 100 kDa, and the MYC-SETDB1-Tudor is about 75 kDa. The results of immunoprecipitation indicated that both deleted fragments interacted with TRAF4 (Figure 5B,C). Notably, both deleted fragments contained the Tudor domain, which suggested to us that TRAF4 might interact with the Tudor domain of SETDB1. To rule out the possibility that other domains of SETDB1 interact with TRAF4, we constructed deletion fragments of the individual domains of SETDB1. One was an MBD domain-only fragment (MBD) and the other was a SET domain-only fragment (SET) (Figure 5D). The molecular weight of MYC-SETDB1-MBD is about 20 kDa, and that of MYC-SETDB1-SET is about 75 kDa. Immunoprecipitation experiments of the deleted fragment confirmed that TRAF4 only interacted with the Tudor domain of SETDB1 (Figure 5E,F). To exclude nonspecific bands, we confirmed the specific band of the deletion fragment (SET) by Western blot (Appendix A).

It has been reported that the Tudor domain is a histone reader [33], and the Tudor domain is important for SETDB1 recognition of methylation sites and protein interactions [34,35]. The results of immunoprecipitation suggested to us that the interaction of TRAF4 with the Tudor domain of SETDB1 may affect the stability and function of SETDB1.

### 2.5. TRAF4 Mediates Atypical Ubiquitination of SETDB1

As in the previous experiments, we found that the protein expression of SETDB1 decreased following TRAF4 knockdown (Figure 3B). We examined the mRNA expression of SETDB1 in glioblastoma cells to exclude the effect of reduced mRNA on SETDB1 protein expression. As shown by qRT-PCR results, the mRNA expression of SETDB1 was not significantly decreased following TRAF4 knockdown (Figure 6A). As an E3 ubiquitin ligase, TRAF4 has been reported to be involved in the ubiquitination of many substrate proteins [15,19]. We performed protein turnover assays and the results suggested that TRAF4 knockdown can effectively reduce the stability of SETDB1 in glioblastoma cells treated with de novo protein synthesis inhibitor cycloheximide (CHX) (Figure 6B,C). Here, we hypothesized that TRAF4 mediated the ubiquitination of SETDB1. To test the hypothesis, we performed ubiquitination experiments of SETDB1. The results indicated that the ubiquitination level of SETDB1 decreased following TRAF4 knockdown, while the ubiquitination level of SETDB1 increased following TRAF4 overexpression (Figure 6D,E). 

Different types of ubiquitin chains lead to different fates for substrate proteins. Ubiquitin may be attached to the substrate through one or more of seven lysine residues (K6, K11, K27, K29, K33, K48, and K63) [18]. It has been reported that TRAF4 functions to mediate K27, K29, and K63-linked ubiquitination of substrate proteins [15,16,19]. To examine the type of ubiquitin linkages mediated by TRAF4 in SETDB1 ubiquitination, we introduced lysine site-mutated ubiquitin in place of wild-type ubiquitin. Ubiquitination experiments suggested that TRAF4 may mediate SETDB1 ubiquitination through the K29 and K33 ubiquitin linkages (Figure 6F). Existing studies suggest that K29-linked ubiquitination may increase the enzymatic activity and stabilize function of substrate proteins [19,36], and it has been reported that K33-linked ubiquitination plays an important role in protein transport, complex formation, and protein interaction [37,38,39]. Our results imply that TRAF4-mediated ubiquitination of SETDB1 may affect the stability and function of the latter.

### 2.6. SETDB1 Overexpression Significantly Restores Cell Proliferation of TRAF4 Knockdown Glioblastoma Cells

To further confirm that TRAF4 promoted glioblastoma proliferation through regulating the stability and function of SETDB1, we overexpressed SETDB1 in TRAF4 knockdown glioblastoma cells (Figure 7B). The results of MTT experiments showed that overexpression of SETDB1 significantly restored cell proliferation (Figure 7A). EdU incorporation experiments also indicated that the proliferative capacity of TRAF4-knockdown glioblastoma cells was significantly increased following SETDB1 overexpression (Figure 7C). Similarly, we examined the intracellular AKT pathway, and the results of Western blot showed that overexpression of SETDB1 in TRAF4-knockdown glioblastoma restored the activation of AKT pathway (Figure 7B). We further examined the growth ability of glioblastoma. The colony formation assays suggested that overexpression of SETDB1 in TRAF4-knockdown glioblastoma partially restored cell growth (Figure 7D). Taken together, our experimental results suggested that overexpression of SETDB1 can restore the proliferation of TRAF4 knockdown glioblastoma cells to a certain extent. These indicated that TRAF4 promoted glioblastoma proliferation, at least in part, by regulating SETDB1 stability and function.

### 2.7. TRAF4 Knockdown Suppresses Tumor Growth in Mice

To explore the role of TRAF4 on glioblastoma growth in vivo, we subcutaneously injected glioblastoma cells in mice to perform animal experiments. The results showed that the tumor formation of glioblastoma was significantly inhibited following TRAF4 knockdown (Figure 8A,B). Through immunohistochemical experiments, we verified that the TRAF4 expression in subcutaneous tumor tissue decreased, and the SETDB1 expression were also decreased in accordance with TRAF4, which were consistent with previous cell experiments (Figure 8C). These results indicated that TRAF4 knockdown suppressed glioblastoma growth in vivo.

Our studies demonstrated that TRAF4 acts as an E3 ubiquitin ligase that interacts with the Tudor domain of SETDB1 and mediates ubiquitination stabilization of the latter. Stably expressed and functional SETDB1 cooperates with PI3K to activate the AKT pathway. The activated AKT pathway ultimately promotes glioblastoma tumorigenesis (Figure 8D).

## 3. Discussion

Ubiquitination plays a key role in tumor progression, regulating and affecting cell growth, metabolism, proliferation, and metastasis [18,40,41,42]. There is increasing evidence that TRAF4, as an E3 ubiquitin ligase, is involved in the growth, proliferation, and metastasis of tumor cells [13,16,28]. Glioblastoma is an extremely malignant tumor with a poor prognosis for patients. The 5-year survival rate for glioblastoma patients is only 5.8% [2]. Clinical data from tumor patients show that TRAF4 is significantly overexpressed in glioblastoma; however, there is little research on the role and regulatory mechanism of TRAF4 in glioblastoma proliferation.

Here, our research suggested that TRAF4 expression was positively correlated with glioma grade and malignant subtype (Figure 1). Knockdown of TRAF4 inhibits the growth, proliferation, migration, and invasion of glioblastoma cells (Figure 2 and Appendix A). In our findings, the effect of TRAF4 on the malignant phenotype of glioblastoma cells is consistent with previous findings [22]. Moving further than previous study, our study characterized TRAF4 in glioblastoma systematically and explored the effect of TRAF4 on the clone formation, self-renewal, and in vivo tumorigenesis of glioblastoma cells. Previous study indicated that TRAF4 is an important factor mediating the activation of the AKT pathway in lung cancer [28]. The AKT pathway plays a crucial role in various intracellular regulation and processes and is abnormally activated in a variety of cancers, including glioblastoma [43]. We found that activation of the AKT pathway is inhibited following TRAF4 knockdown in glioblastoma cells (Figure 3). Examining the AKT pathway by Western blot and immunoprecipitation, we find that TRAF4 interacts with SETDB1 to maintain the stability of the latter (Figure 4, Figure 5 and Figure 6). Stable SETDB1 expression is important for AKT pathway activation [30,31], and loss of TRAF4 reduces SETDB1 stabilization and thus suppresses intracellular AKT pathway activation. The AKT pathway is also a very important regulator of cancer stem cells, and the combined utilization of pathway inhibitors targeting the AKT pathway with other cancer treatments is the most effective strategy for cancer treatment [43,44]. Our studies indicate that targeting TRAF4 to inhibit activation of the AKT pathway is meaningful (Figure 3). Collectively, our results enrich the mechanism by which TRAF4 mediates AKT pathway activation, providing insights for AKT pathway research and related cancer treatments.

A large number of studies on post-translational modifications and epigenetic regulation have deepened the understanding of the occurrence and progression of many diseases, especially cancer, and also promoted the progress of early diagnosis and treatment for cancer medicine [45,46,47]. Recent studies have found that SETDB1 mediates AKT methylation and promotes membrane recruitment, phosphorylation, and activation of AKT [30,31]. Our study explored the upstream of the SETDB1-AKT pathway and revealed the mechanism by which TRAF4 regulates the SETDB1-AKT pathway. Protein methylation is an interesting and promising direction. In the cytoplasm, the mechanism by which TRAF4 regulates SETDB1-mediated protein methylation remains to be studied. SETDB1 plays an important role in tumor progression, participates in many tumor processes, and is a potential target for tumor therapy [48]. Moreover, recent studies have shown that SETDB1 is a promising new target for tumor immunotherapy [49,50]. We found that TRAF4 regulates the stability and function of SETDB1 through ubiquitination (Figure 6 and Figure 7). Given the important role of SETDB1 in tumor immune evasion, this provides new ideas and directions for targeting SETDB1. What role TRAF4 plays in tumor immune evasion requires further exploration by subsequent researchers. 

Overall, our results reveal novel mechanisms by which TRAF4 is involved in regulating the AKT pathway, suggesting that TRAF4 may serve as a biomarker for glioblastoma and a new target for cancer therapy. 

## 4. Materials and Methods

### 4.1. Cell Culture

Human glioblastoma cell lines (A172, LN-229, U87-MG, U118-MG, and U251-MG) and normal astroglia cells (SVGP12) were obtained from American Type Culture Collection (ATCC: Manassas, VA, USA). Cells were cultured in Dulbecco’s Modified Eagle’s Medium (DMEM) (Gibco: Grand Island, NY, USA) with 10% fetal bovine serum (Vivacell: Chongqing, China) and 1% penicillin–streptomycin solution in 37 °C and 5% CO_2_ incubator.

### 4.2. Antibodies and Reagents

Antibodies used in this article include anti-TRAF4 (Proteintech: Wuhan, China, CAS: 66755-1-Ig), anti-TRAF4 (Abcam: NY, USA, CAS: ab245666), anti-Tubulin (Proteintech: Wuhan, China, CAS: 66031-1-Ig), anti-E-cadherin (Proteintech: Wuhan, China, CAS: 20874-1-AP), anti-MMP2 (Proteintech: Wuhan, China, CAS: 10373-2-AP), anti-β-catenin (Cell Signaling Technology: Danvers, MA, USA, CAS: 8480), anti-CCND1 (Proteintech: Wuhan, China, CAS: 60186-1-Ig), anti-GSK3β (Proteintech: Wuhan, China, CAS: 22104-1-AP), anti-p-GSK3β (Cell Signaling Technology: Danvers, MA, USA, CAS: 5558), anti-AKT1 (Proteintech: Wuhan, China, CAS: 60203-2-Ig), anti-p-AKT (Cell Signaling Technology: Danvers, MA, USA, CAS: 13038), anti-SETDB1(Cell Signaling Technology: Danvers, MA, USA, CAS: 2196), anti-Flag (Proteintech: Wuhan, China, CAS: 66008-3-Ig), anti-Flag (Proteintech: Wuhan, China, CAS: 80010-1-RR), anti-MYC-tag (Proteintech: Wuhan, China, CAS: 60003-2-Ig), anti-MYC-tag (Proteintech: Wuhan, China, CAS: 16286-1-AP), and anti-HA (Proteintech: Wuhan, China, CAS: 51064-2-AP). 

Reagents used in this article including Cycloheximide (CHX) (Merck: Kenilworth, NJ, USA, CAS: 66-81-9) and proteasome inhibitor (MG-132) (MCE: Chongqing, China, CAS: 133407-82-6).

### 4.3. Tumor Database Analysis

The gene expression data of TRAF4 was downloaded and analyzed from the CGGA database (http://www.cgga.org.cn/ (accessed on 30 June 2022)), GEPIA database (http://gepia.cancer-pku.cn/), GlioVis database (http://gliovis.bioinfo.cnio.es/ (accessed on 30 June 2022)), and TCGA database (https://www.cancer.gov/nci/organization/ccg/research/structural-genomics/tcga (accessed on 30 June 2022)). Cutoff separating was based on the expression level of TRAF4, Kaplan–Meier analysis and gene correlation were plotted by CGGA, GlioVis, TCGA. We performed protein–protein interaction analysis in the online tumor databases GeneMANIA (https://genemania.org (accessed on 30 June 2022)) and String (https://stringdb.org (accessed on 30 June 2022)).

### 4.4. Transfection and Infection

shTRAF4 sequences were recombined into pLKO.1-puro plasmid, and the control was pLKO.1-puro empty plasmid. The sequences are listed below:

shTRAF4#1: GTATGGCCTAGATGTTTCATA

shTRAF4#2: GAGAGTGTCTACTGTGAGAAT

Flag-tagged full length of TRAF4 recombined into the pCDH-CMV-MCS-EF1-Puro plasmid (Youbio: Wuhan, China). MYC-tagged full length and various deletion fragments recombined into the Pcdh-CMV-MCS-EF1-Puro plasmid (Genecreate: Chongqing, China).

For transfection, lipofectamine2000 (Thermo: Waltham, MA, USA, CAS:11668019) and specific constructs were mixed and added into the 6-well plate. The medium was changed after 6–8 h. Cells were retrieved after 40–48 h. 

For infection, lentivirus was produced by HEK-293 FT cells. After two times of infection, the medium was refreshed and puromycin was added to screen the cells that are infected [23]. 

### 4.5. Western Blot Analysis 

The recovered cells were lysed with RIPA buffer (Beyotime: Shanghai, China) for 30–60 min, and the supernatants were collected following centrifuge 12,000 rpm at 4 °C for 10 min. The protein mixed with loading buffer was denatured at 96 °C, followed by SDS-PAGE and transmembrane (Bio-Rad: Hercules, CA, USA). We incubated the primary antibody overnight at 4 °C and incubated the secondary antibody for 1–2 h at room temperature. The PVDF membrane was exposed in the gel imaging system (Qinxiang: Shanghai, China).

### 4.6. Immunoprecipitation

The recovered cells were lysed with IP-lysis buffer (Beyotime: China) for 30–60 min, and the supernatants were collected following centrifuge 12,000 rpm at 4 °C for 10 min. The specific antibody or IgG was added to the cell lysate and incubated overnight, followed by the addition of Protein A + G Agarose (Beyotime: China) for more than 4 h. Agarose was washed five times with PBS, then mixed with loading buffer and denatured at 96 °C. Samples were examined by Western blot.

### 4.7. Quantitative Real-Time PCR (qRT-PCR)

Total RNA was extracted using standard TRIzol methods, and mRNA was subsequently converted to cDNA using Reverse Transcription Kit (Promega: Shanghai, China). SYBR qPCR SuperMix Plus was used for qRT-PCR (Novoprotein: Shanghai, China). The cDNA was used to perform qRT-PCR. β-actin was used to normalize mRNA input. RNA expression levels were calculated according to the comparative Ct method (ΔΔCT). The sequence of the primers is listed in Table 1.

### 4.8. Ubiquitination Assay

MYC-tagged SETDB1, HA-tagged Ub, and shTRAF4/Flag-tagged TRAF4 constructs were cotransfected into 293FT cells. The cells were treated with MG-132 6–8 h before collecting, and the rest of the steps were the same as in the immunoprecipitation experiments.

### 4.9. Protein Turnover Assay

Treatment of stable cell lines was carried out with CHX at a concentration of 100 μg/mL, and cells were collected in a time gradient. Cells were lysed for Western blotting.

The grey values of Western blots were calculated with ImageJ to obtain protein turnover, and the results are presented graphically.

### 4.10. Immunohistochemistry (IHC)

For immunohistochemistry, tissue sections were successively deparaffinized, rehydrated, antigen retrieved in citrate buffer, endogenous peroxidase inactivated with 3% hydrogen peroxide, and blocked with goat serum. After incubation with specific primary antibodies overnight at 4 °C, tissue sections were incubated with horseradish peroxidase-conjugated secondary antibodies. Antibodies were visualized using diaminobenzidine treatment and nuclei were counterstained with hematoxylin.

### 4.11. MTT and EdU Assay

For MTT assay, 1 × 10^3^ cells were seeded into the 96-well plate. The cells were grouped in a time gradient and treated with MTT for 2 h, then the medium was removed and 200 μL DMSO was added for incubation. The 560 nm absorbency was measured by microplate reader.

For EdU assay, 2 × 10^4^ cells were seeded into the 24-well plate. We used the EdU-488 Cell Proliferation Detection Kit (Beyotime: China) for experiments.

### 4.12. Wound-Healing Assay and Transwell Assay

For wound-healing assay, 2 × 10^5^ cells were seeded into the 6-well plate and cultured until the cell density reached 80%. The cells were starved overnight with a change of serum-free medium, and wounds were made on the cells using a pipette tip.

For Transwell assay, 5 × 10^4^ cells were seeded into Transwell chamber (Corning: USA) with 200 µL serum-free medium, and 500 µL normal medium was added into the 24-well plate. After about 8 h, the cells on the Transwell chamber were fixed with paraformaldehyde solution, stained with crystal violet, and finally the cells on the upper membrane of the Transwell chamber were wiped off. Stained cells in the lower membrane of the Transwell chamber were examined by microscope. In invasion experiments, Matrigel (R&D Systems: USA) was additionally added into the Transwell chamber.

### 4.13. Colony Formation and Soft Agar Assay

For colony formation, 2 × 10^3^ cells were seeded into the 6-well plate and cultured for 4–7 days. Cells were fixed with paraformaldehyde solution for 20 min and stained with crystal violet for 30 min.

Then, 1.2% Agarose and 2 × DMEM medium were mixed 1:1 by volume and added into the 6-well plate as bottom gel. Then, 1.2% Agarose, 2 × DMEM medium, and cell suspension (1 × 10^3^ cells) were mixed 1:1:2 by volume and added into the 6-well plate as upper gel. Cells were cultured in an incubator for about 4 weeks. We examined the clones under the microscope. Cells were stained with MTT.

### 4.14. Gene Set Enrichment Analysis (GSEA)

The CGGA database was downloaded from the Chinese Glioma Genome Atlas (http://www.cgga.org.cn/ (accessed on 30 June 2022)). The gene sets were obtained from the Molecular Signatures Database (MsigDB, http://software.broadinstitute.org/gsea/index.jsp (accessed on 30 June 2022)) [23]. The database was analyzed using GSEA (version 4.0.3) for gene sets associated with TRAF4 expression.

### 4.15. Animal Experiments

A total of 12 mice were randomly divided into two groups; one was the U87-MG group (6 mice) and the other was the LN-229 group (6 mice). The cells of the control and the experimental group were infected with the empty construct and the shTRAF4 construct, respectively. We collected glioblastoma cells and resuspended them in PBS. The control and the experimental cells were injected subcutaneously into the right and left hind limbs of mice, respectively. A total of 1×10^5^ cells were injected per site. Tumors grew to an appropriate size according to animal ethics. Mice were sacrificed 25 days post glioblastoma cells injection and subcutaneous tumors were weighed.

### 4.16. Animal Ethics

Four-week-old female NOD/SCID mice (Beijing Animal Research Center, Beijing, China) were purchased and housed in our SPF room. The survival status of the mice was observed every day. We changed the mice’s feed, sterile drinking water, and bedding frequently to ensure that the mice lived in a clean environment. In animal experiments, subcutaneous glioblastoma xenografts were not allowed to grow for more than four weeks, tumor weight in mice was not more than 10% of the mouse body weight, and the average tumor diameter was not more than 15 mm. Isoflurane anesthesia was used to reduce the pain of the mice when the tumors were harvested. Mice cadavers were stored at −20 °C and then incinerated by Laibite Biotech Inc. (Chongqing, China). All animal studies were approved by the Institutional Animal Care and Use Committee of Southwest University.

### 4.17. Statistical Process

Each of the above experiments was performed independently at least three times. Statistics in the experiments were analyzed by GraphPad Prism 7.0. All data in this study were analyzed and shown as mean ± SD. The two-tailed Student’s *t* test was used to analyze the two groups’ data. The data were confirmed to be significant if the *p*-value < 0.05; * *p* < 0.05, ** *p* < 0.01, *** *p* < 0.001. 

## 5. Conclusions

Taken together, our study characterizes TRAF4 in glioblastoma and demonstrates that the E3 ubiquitin ligase TRAF4 can activate the AKT pathway by stabilizing the expression of SETDB1, ultimately promoting glioblastoma proliferation. Our research elucidates a novel mechanism showing how TRAF4 is involved in glioblastoma proliferation and furthers our understanding of the role of TRAF4 ubiquitination. Given the important role of SETDB1 in cancer treatment [50], our study suggests possible potential research and therapeutic directions.

## Figures and Tables

**Figure 1 ijms-23-10161-f001:**
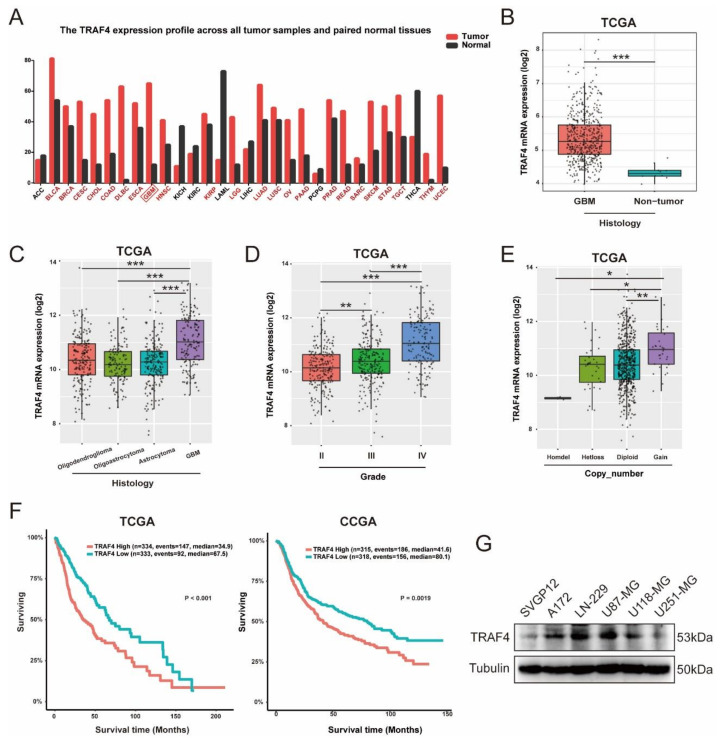
TRAF4 is highly expressed in glioblastoma and predicts poor prognosis. (**A**) Expression of TRAF4 in tumor samples and paired normal tissues was obtained from the GEPIA database. (**B**) Expression of TRAF4 in glioblastoma and non-tumor was obtained from the TCGA database. Box plot of TRAF4 expression with the log-rank test *p*-values indicated. (**C**–**E**) Box plot of TRAF4 expression levels by grade, histological subtype, and copy number in glioma set with log-rank test *p*-values indicated. (**F**) The TCGA and CGGA glioma databases were used to examine the correlation of TRAF4 expression with patient prognosis. (**G**) TRAF4 expression in normal astroglia cell (SVGP12) and glioblastoma cell lines (A172, LN-229, U87-MG, U118-MG, and U251-MG) were detected by Western blotting. All data represent the mean ± SD from at least three independent experiments. Student’s *t* test was performed to analyze significance; * *p* < 0.05, ** *p* < 0.01, *** *p* < 0.001.

**Figure 2 ijms-23-10161-f002:**
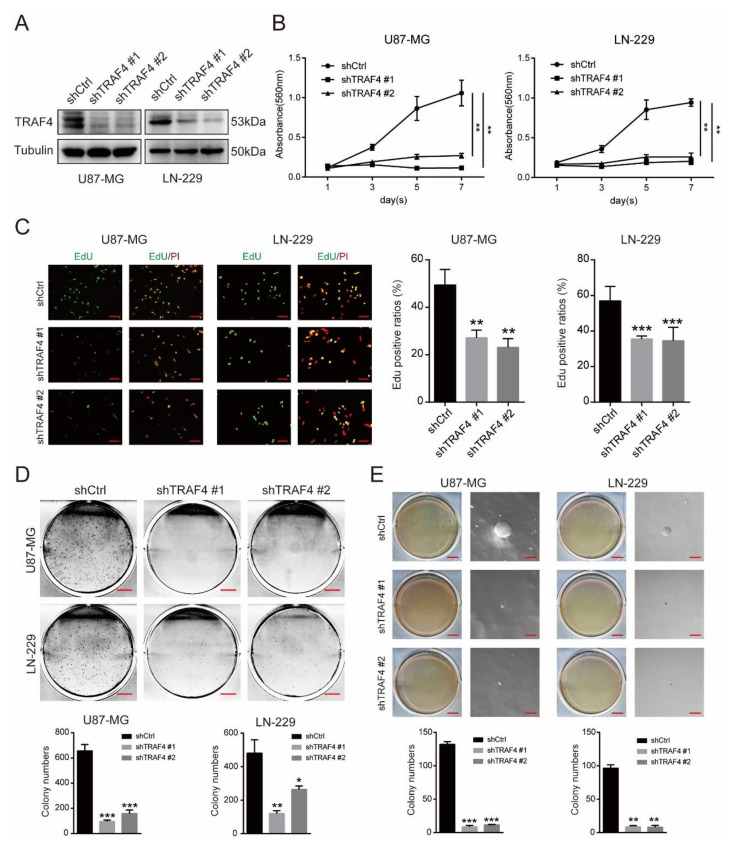
Downregulation of TRAF4 in glioblastoma cells inhibits cell proliferation. (**A**) TRAF4 expression was detected by Western blotting in glioblastoma cells following TRAF4 knockdown. (**B**,**C**) The viability and proliferation of glioblastoma cells detected by MTT assay and EdU following TRAF4 knockdown. Scale bars = 20 µm. (**D**,**E**) The clone-forming and self-renewal ability of glioblastoma cells detected by colony formation and soft agar assay following TRAF4 knockdown. Scale bars (original graphic) = 5 mm, scale bars (magnified graphic) = 100 µm. All data represent the mean ± SD from at least three independent experiments. Student’s *t* test was performed to analyze significance; * *p* < 0.05, ** *p* < 0.01, *** *p* < 0.001.

**Figure 3 ijms-23-10161-f003:**
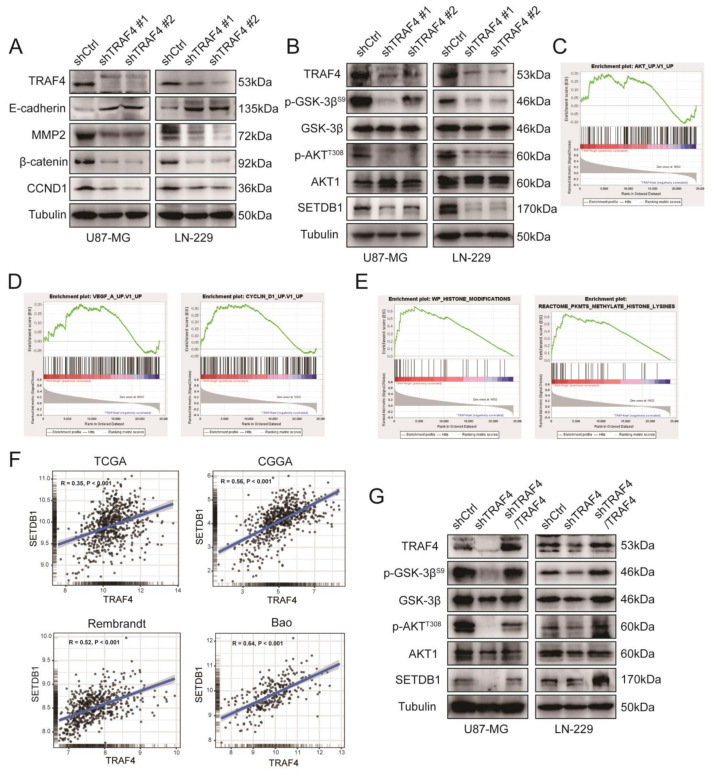
Downregulation of TRAF4 in glioblastoma cells inhibits cellular AKT pathway activation, whereas restoration of TRAF4 expression counteracts the inhibition. (**A**) The expression of glioblastoma cell growth and metastasis-related proteins was detected by Western blotting following TRAF4 knockdown. (**B**) AKT pathway activation-related protein expression in glioblastoma cells were detected following TRAF4 knockdown. (**C**) In GSEA, enrichment analysis of TRAF4 and AKT pathway gene sets. (**D**) Enrichment analysis of TRAF4 and cell metastasis-related gene sets and CCND1-related gene sets. (**E**) Enrichment analysis of TRAF4 and histone modification gene sets and histone lysine methylation gene sets. (**F**) Gene correlation analysis of TRAF4 with SETDB1 in glioma databases TCGA, CGGA, Rembrandt and Bao. (**G**) AKT pathway activation-related proteins expression was detected by Western blotting following TRAF4 knockdown and then overexpressed. Student’s *t* test was performed to analyze significance.

**Figure 4 ijms-23-10161-f004:**
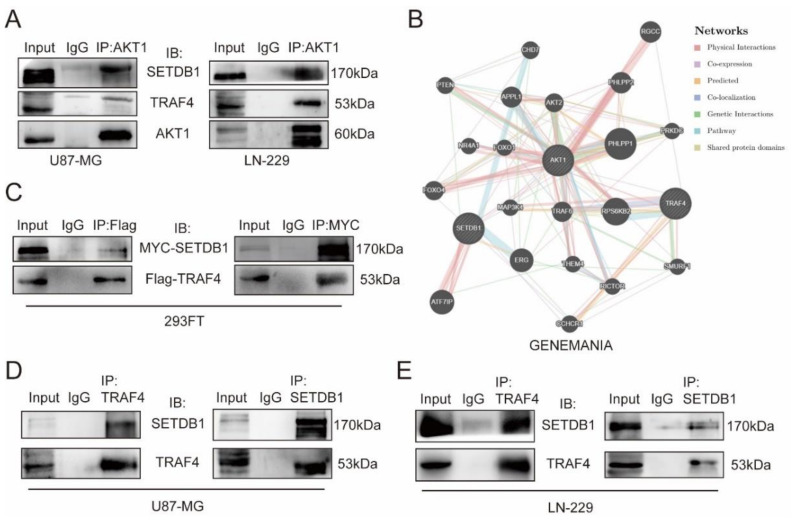
TRAF4 interacts with SETDB1. (**A**) Whole-cell extracts (WCE) of U87-MG and LN-229 cells were collected and subjected to immunoprecipitation assays (AKT1 for IP) and immunoblotting. The interaction of AKT1 with TRAF4 and SETDB1 in glioblastoma cells was examined. (**B**) Protein–protein interaction (PPI) analysis in the online tumor databases GeneMANIA. (**C**) 293FT cells infected with Flag-TRAF4 and MYC-SETDB1 were immunoprecipitated with anti-Flag antibody/anti-MYC antibody, followed by immunoblotting. The interaction of exogenously expressed Flag-TRAF4 and MYC-SETDB1 in 293FT was examined. (**D**,**E**) Whole-cell extracts (WCE) of (**D**) U87-MG and (**E**) LN-229 cells were collected and subjected to immunoprecipitation assays (TRAF4/SETDB1 for IP) and immunoblotting. The interaction of TRAF4 and SETDB1 in glioblastoma cells was examined.

**Figure 5 ijms-23-10161-f005:**
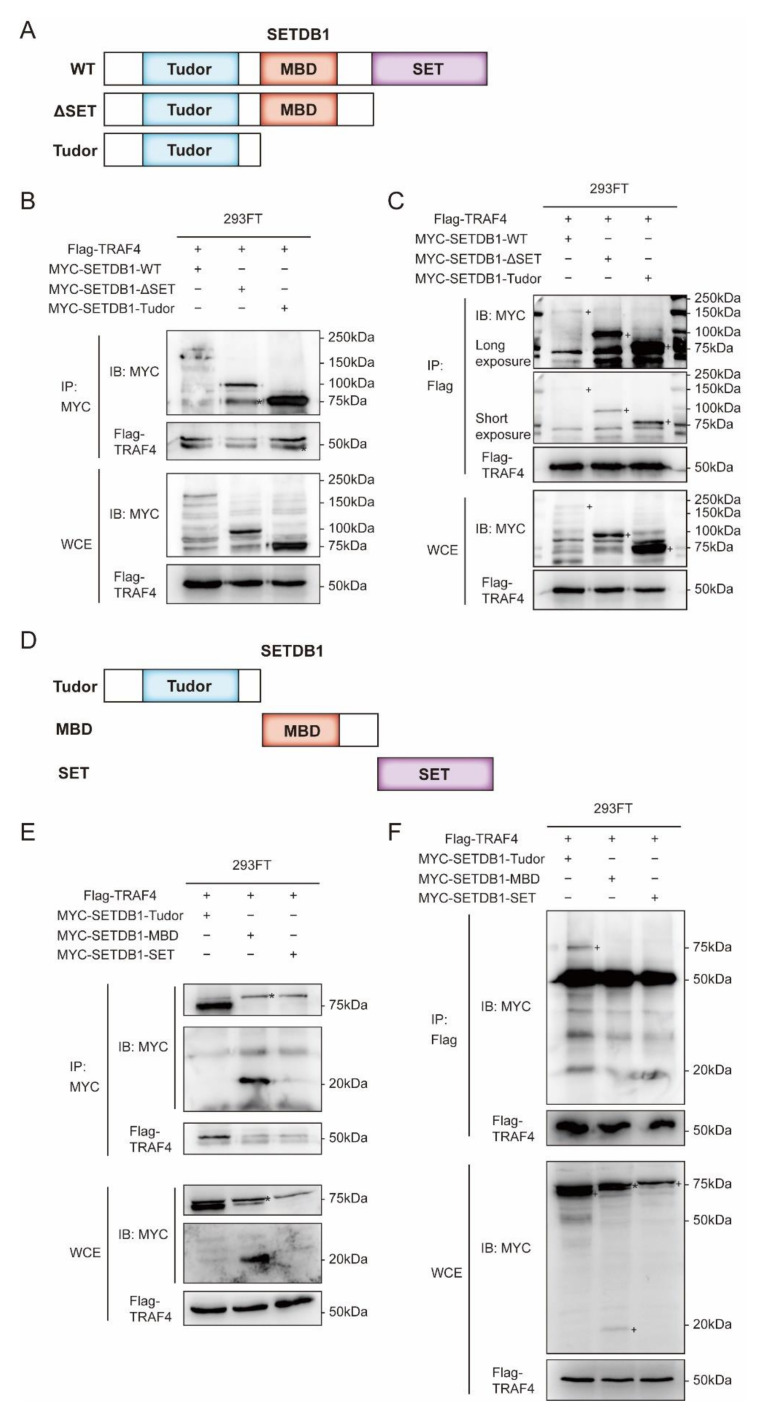
TRAF4 only interacts with the Tudor domain of SETDB1. (**A**) Schematic representation of full-length and deleted fragments of SETDB1. (**B**,**C**) 293FT cells infected with Flag-TRAF4 and MYC-SETDB1-WT/MYC-SETDB1-ΔSET/MYC-SETDB1-Tudor were immunoprecipitated with (**B**) anti-MYC antibody/(**C**) anti-Flag antibody, followed by immunoblotting. The interaction of TRAF4 with deleted fragments of SETDB1 (ΔSET and Tudor) were examined. (**D**) Schematic representation of deleted fragments of SETDB1. (**E**,**F**) 293FT cells infected with Flag-TRAF4 and MYC-SETDB1-Tudor/MYC-SETDB1-MBD/MYC-SETDB1-SET were immunoprecipitated with (**E**) anti-MYC antibody/(**F**) anti-Flag antibody, followed by immunoblotting. The interaction of TRAF4 with deleted fragments of SETDB1 (Tudor, MBD, and SET) were examined; + positive signal, * non-specific signal.

**Figure 6 ijms-23-10161-f006:**
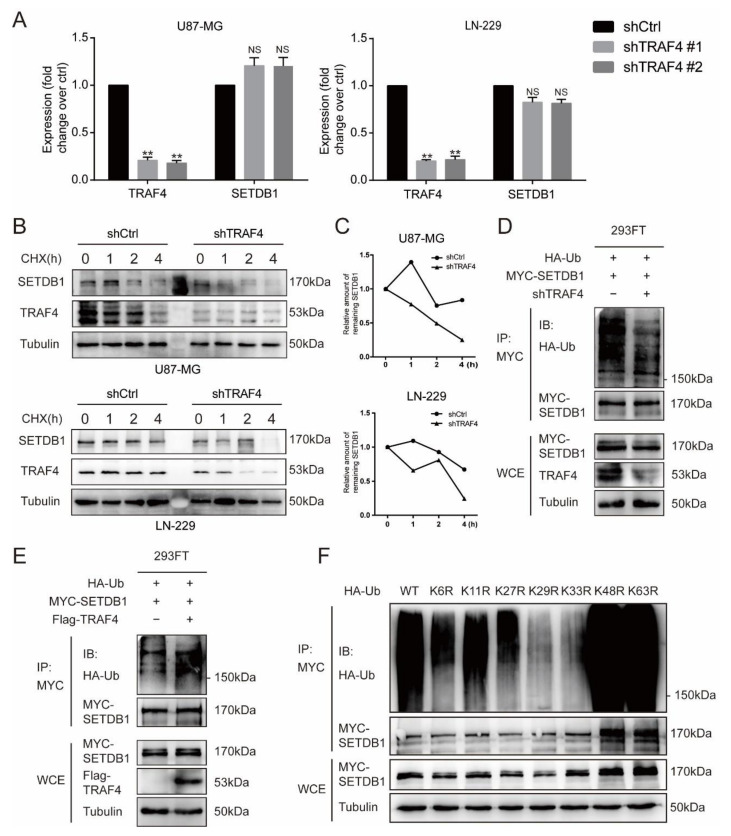
TRAF4 regulates SETDB1 stabilization and mediates atypical ubiquitination of SETDB1. (**A**) TRAF4 and SETDB1 mRNA levels were examined by qRT-PCR in glioblastoma cells. (**B**,**C**) Control group and TRAF4 knockdown cells were treated with CHX in a time gradient (0 h, 1 h, 2 h, 4 h). The protein expression of SETDB1 was detected by Western blot. The grey values were calculated by ImageJ. (**D**,**E**) MYC-tagged SETDB1, HA-tagged Ub, and (**D**) shTRAF4/ (**E**) Flag-tagged TRAF4 constructs were cotransfected into 293FT cells. The ubiquitinated SETDB1 proteins were pulled down with anti-MYC antibody and immunoblotted with anti-HA antibody. (**F**) The Flag-tagged TRAF4, MYC-tagged SETDB1, HA-tagged Ub, and ubiquitin mutant constructs were cotransfected into 293FT cells. The 293FT cells were treated with MG132 for ubiquitination assays. Student’s *t* test was performed to analyze significance; ** *p* < 0.01, NS: not significant.

**Figure 7 ijms-23-10161-f007:**
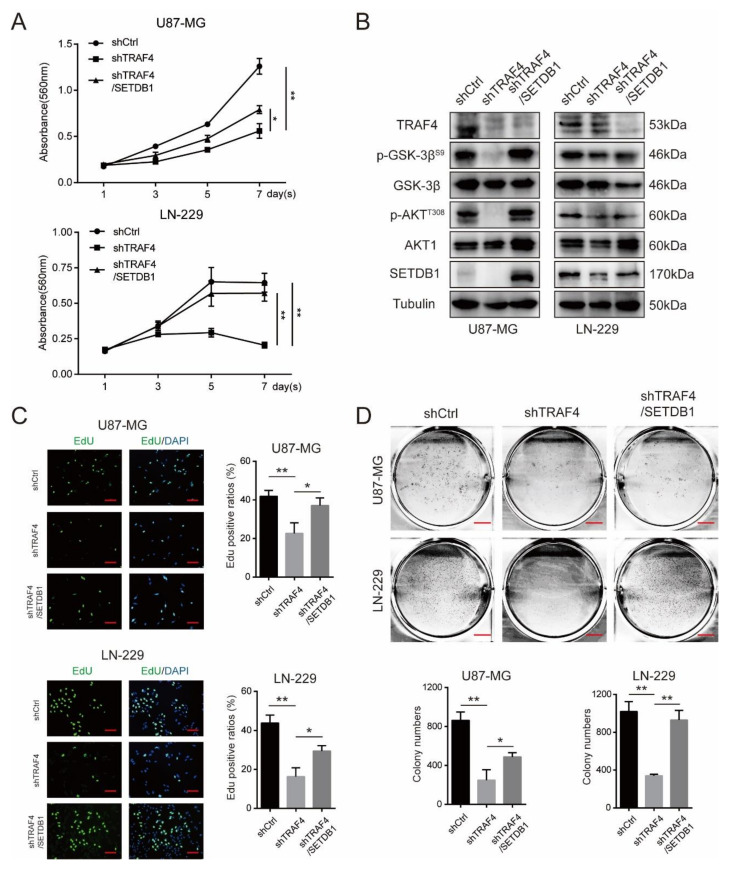
Overexpression of SETDB1 in TRAF4 knockdown glioblastoma cells partially restores cell proliferation and AKT pathway activation. (**A**) The viability of TRAF4 knockdown glioblastoma cells was examined by MTT assay following SETDB1 overexpression. (**B**) AKT pathway activation-related protein expression in TRAF4 knockdown glioblastoma cells were detected following SETDB1 overexpression. (**C**) The proliferation of TRAF4 knockdown glioblastoma cells was examined by EdU assay following SETDB1 overexpression. Scale bars = 20 µm. (**D**) The clone-forming ability of TRAF4 knockdown glioblastoma cells detected by colony formation assay following SETDB1 overexpression. Scale bars = 5 mm. All data represent the mean ± SD from at least three independent experiments. Student’s *t* test was performed to analyze significance; * *p* < 0.05, ** *p* < 0.01.

**Figure 8 ijms-23-10161-f008:**
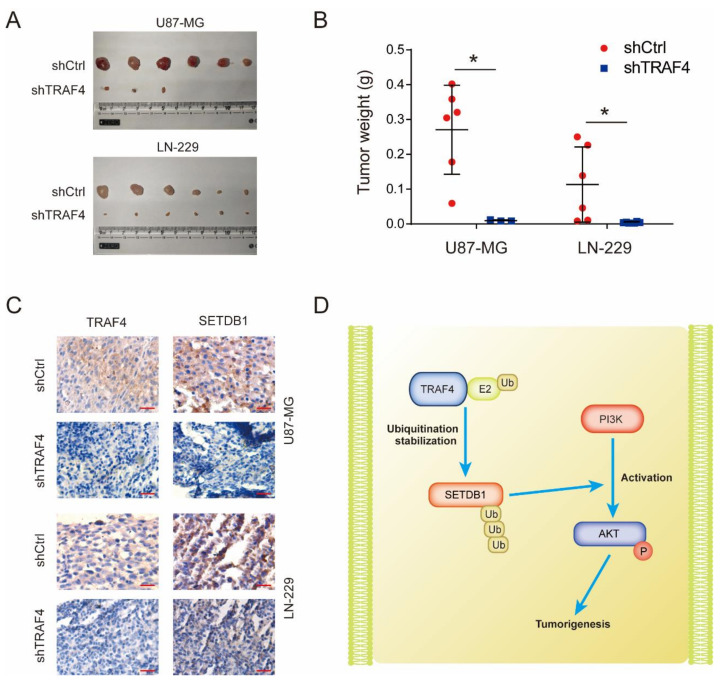
Knockdown of TRAF4 inhibits tumor formation of glioblastoma cells in vivo. (**A**) The in vivo tumorigenic ability of glioblastoma cells was detected by xenograft assay in mice following TRAF4 knockdown. (**B**) Tumor tissue mass in xenograft assay. (**C**) TRAF4 and SETDB1 expression in mice subcutaneous tumor tissues were examined by immunohistochemistry. Scale bars = 20 µm. (**D**) Overview of TRAF4 stabilizing SETDB1 by ubiquitination to promote AKT pathway activation, thereby promoting tumorigenesis. Student’s *t* test was performed to analyze significance; * *p* < 0.05.

**Table 1 ijms-23-10161-t001:** Primer pairs for real-time PCR.

Primer Pairs for Real-Time PCR	
β-actin-F	CATGTACGTTGCTATCCAGGC
β-actin-R	CTCCTTAATGTCACGCACGAT
TRAF4-F	TATTGGGCCTGCCTATCCG
TRAF4-R	CAAAACTCGCACTTGAGGCG
SETDB1-F	AGGAACTTCGGCATTTCATCG
SETDB1-R	TGTCCCGGTATTGTAGTCCCA

## Data Availability

The data presented in this study are available on request from the corresponding author.

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
