# Peer review of "TRAF4 Promotes the Proliferation of Glioblastoma by Stabilizing SETDB1 to Activate the AKT Pathway"

_ijms, 2022, doi:10.3390/ijms231710161_

Round 1
Reviewer 1 Report
In the manuscript „TRAF4 promotes the proliferation of glioblastoma by stabilizing SETDB1 to activate the AKT pathway” Gu et al. explore the role of TRAF4 – SETDB1 – AKT axis in glioblastoma proliferation, migration and invasiveness. The results are original and interesting, and generally the quality of experimental data and their presentation is high. However, there are some technical issues to be addressed to improve the manuscript and some results to be more extensively described or commented/discussed.
MAJOR:
1. What is the subcellular localization of TRAF4 and SETDB1 proteins in GBM cell lines?
2. Figure 6 C and D. Why has anti-SETDB1 antibody been used for the detection of overexpressed myc-SETDB1? Here, anti-myc antibody should be applied.
3. There is discrepancy between results shown in Figures 5 and 6. MYC-WT SETDB1 (full length) is very poorly detected when FLAG TRAF4 is co-expressed (Figure 5 B and C). On the other hand, the band is clearly visible in Figures 6 C, D and E. Full length SETDB1 is expected to pull down on FLAG TRAF4, since TUDOR domain is present.
4. In experiments involving CHX, the measurement of band density from at least 3 independent experiments, quantification and graphical presentation is very much desired.
MINOR:
1. The visibility of colonies formed by GBM cells is very poor (Figures 2 and 7)
2. Molecular weight of proteins should be indicated along with each Western blot picture.
3. Figure 3F presents data from TCGA, showing the correlation between TRAF4 and SETDB1 expression at the mRNA level (?). Though, RT-PCR data in Figure 6A indicate that TRAF4 knock-down doesn’t affect SETDB1 expression. Please, comment on that two observations, as they are somewhat contrary.
4. Please provide full description of experiments depicted in Figure 4, especially which antibody was used for immunoprecipitation, and which one for Western blot. The information on cell infection (plasmids, time), if applicable, should be provided as well.
5. The expected molecular weight of deletion mutants of SETDB1 and constructs containing one protein domain of SETDB1 should be provided (Figure 5). What is the localization of myc tag?
6. Whenever antibody recognizing phosphorylated protein is used, the phosphorylated amino acid should be specified. There are multiple phosphorylation sites at GSK3b and Akt proteins, and they have different function, e.g. only specific site on GSK3b is phosphorylated by Akt.
7. The method of gene expression measurement is not provided (SYBR green?)
8. Please separate Edu staining and IHC protocols in M&M section. There’s no reason to put them together.
9. Figure S3: Which band does represent SET domain of SETDB1? Please identify it with arrow adjacent to the blot. Does lower band (50kD) represent heavy chain of IgG used for IP?
10. For how long GBM xenografts were allowed to grow in mice? Please provide this information in M&M section.
Reviewer 2 Report
In this research article, Gu and colleagues have shown that TRAF4 positively correlates with glioma level and malignancy and negatively regulates AKT pathway. Based on their present findings, the authors suggested TRAF4 as a potential biomarker and target for glioblastoma. This study possesses a worth finding out a novel mechanism of TRAF4-mediated regulation of AKT pathway. The manuscript is well written and contains authors’ scientific messages well with ample data supporting current results. I have some additional comments.
The effect of TRAF4 on glioblastoma formation is shown in Figure 8A-C. If the procedure to determine in vivo ability for the TRAF4’s tumor progression hasn’t been fully described in the text, include it in the Methods.
The explanation of Figure 8D would need to be more added in the legends or text. The description of PIP3 doesn’t appear to be included anywhere in the text.
Fig. 5D was not cited in the text. Or Fig. 4D needs to be corrected to Fig. 5D in the second paragraph of 2.4. of page 4.
Some of abbreviated terms (e.g., SETDB1…) were not spelled out. Spell them out when they come first in the text.
It is recommended to modify the tense of the subtitles in Results, and Figure legends, from past to present.
Round 2
Reviewer 1 Report
The most points of the revisions have been adequately addressed, and the figures corrected accordingly. However, the description of in vivo procedures is not entirely clear. Have all animals been sacrificed at the same time point, 4 weeks post GBM cells injection? If yes, this information need to be placed in the manuscript. If some of them were sacrificed earlier, because tumor was too large according to the ethical comitee guidelines, this should be clearly stated, in a supplementary table form.
Whenever cells were infected with virus carrying shRNA or recombinant protein, the term "plasmid" need to be replaced with "construct" or "virus" (e.g. in the description of animal procedures).
There are still numerous typos and grammar errors in the text. I strongly suggest language correction. Definitely, the scientific content deserves better presentation.
